# Growing Winning Subnetworks, Not Pruning Them: A Paradigm for Density Discovery in Sparse Neural Networks

## Abstract

The lottery ticket hypothesis suggests that dense networks contain sparse subnetworks that can be trained in isolation to match full-model performance. Existing approaches—iterative pruning, dynamic sparse training, and pruning at initialization—either incur heavy retraining costs or assume the target density is fixed in advance. We introduce Path Weight Magnitude Product-biased Random growth (PWMPR), a constructive sparse-to-dense training paradigm that grows networks rather than pruning them, while automatically discovering their operating density. Starting from a sparse seed, PWMPR adds edges guided by path-kernel-inspired scores, mitigates bottlenecks via randomization, and stops when a logistic-fit rule detects plateauing accuracy. Experiments on CIFAR, TinyImageNet, and ImageNet show that PWMPR approaches the performance of IMP-derived lottery tickets—though at higher density—at substantially lower cost ( 1.5x dense vs. 3-4x for IMP). These results establish growth-based density discovery as a promising paradigm that complements pruning and dynamic sparsity.

## 1 Introduction

Artificial neural networks (ANNs) power state-of-the-art systems in vision, language, and many other domains. Their success has largely been driven by scaling: larger and denser models trained on larger datasets consistently improve performance (Kaplan et al., 2020). Yet this progress comes at immense computational cost, motivating the search for sparse alternatives that retain dense-level accuracy with reduced training and inference requirements.

The lottery ticket hypothesis (LTH) (Frankle & Carbin, 2018) crystallized this challenge: dense networks contain sparse subnetworks ("winning tickets") that can be trained in isolation to match the full model's accuracy. Iterative Magnitude Pruning (IMP) demonstrates such subnetworks, but at prohibitive cost—often 3-4x more than dense training. This sparked extensive research into pruning-at-initialization (Lee et al., 2018; Tanaka et al., 2020), dynamic sparse training (Mocanu et al., 2018; Evci et al., 2020), and reparameterization methods (Mostafa & Wang, 2019; Kusupati et al., 2020).

Despite their diversity, these approaches share a critical limitation: they assume the density is known or fixed in advance. PaI methods require the user to set a target density from the start; DST maintains a predetermined sparsity budget throughout training; Reparameterization methods control target density either directly(Mostafa & Wang, 2019) or through proxy hyperparameters(Kusupati et al., 2020). Yet in practice, the relationship between density and accuracy is unknown, and this assumption constrains both scientific understanding and practical deployment.

We argue that the ability to discover the density automatically is not a side problem but a central open question in sparse learning. To address it, we propose a shift in paradigm: rather than *destructively pruning* or *preserving* a fixed density, we explore constructive sparse-to-dense growth.

Our method, *Path Weight Magnitude Product-biased Random growth (PWMPR)*, embodies this paradigm. Starting from a sparse seed network, PWMPR grows new connections during training, guided by a topological score rooted in path kernel analysis. Randomization mitigates bottlenecks, and a lightweight logistic-fit rule stops growth once accuracy gains plateau. In this way, PWMPR

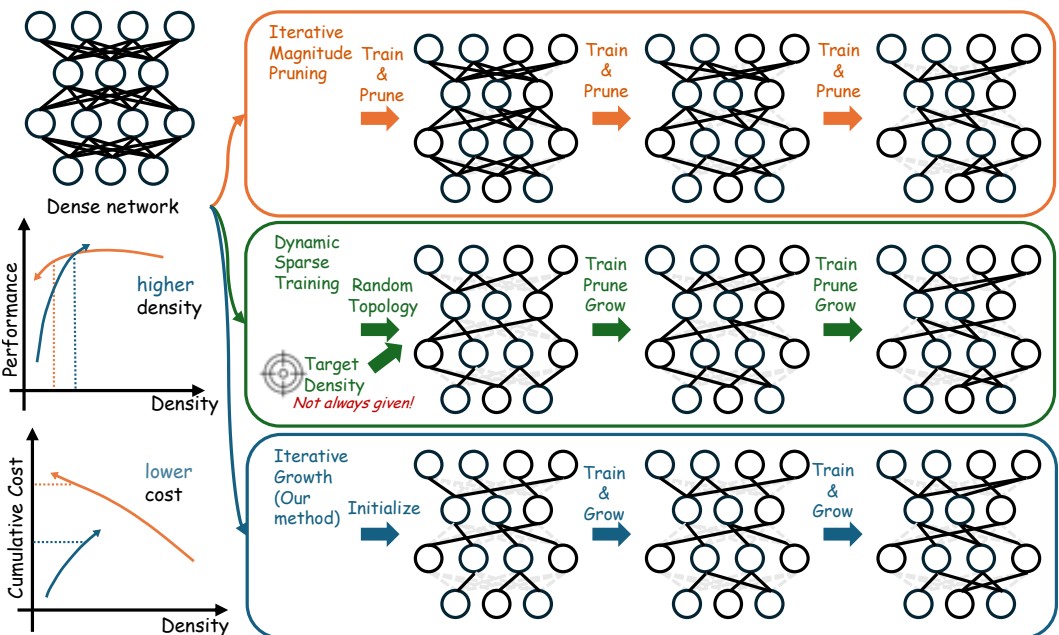

Figure 1: Conceptual comparison of strategies for identifying sparse neural networks that match dense network performance. **Blue**: Iterative Magnitude Pruning (IMP) removes low-magnitude weights after repeated full training cycles. **Orange**: Dynamic Sparse Training (DST) starts from a random sparse topology and alternates pruning and regrowth at a fixed target density. **Green**: Our iterative growth method begins with a sparse network and progressively adds connections during training. The performance-density and cost-density sketches highlight the contrasting trade-offs between pruning and growth.

constructs subnetworks and discovers their density simultaneously, without requiring dense pretraining or preset sparsity.

As illustrated in Figure 1, this growth-based view complements pruning and DST. While PWMPR typically reaches higher densities than IMP-derived tickets, it does so at much lower training cost ( 1.5x dense vs. 3-4x for IMP-C), producing subnetworks that approach lottery-ticket performance efficiently. We see this as a first step in establishing growth-based density discovery as a new paradigm in sparse neural network training, with implications for hybrid grow-prune algorithms, transformer-specific growth rules, and broader theoretical analysis.

## 2 PROBLEM STATEMENT

Training sparse neural networks efficiently requires balancing three factors: how small the network is (its density), how well it performs on the task, and how much training it costs. IMP starts from a dense model, repeatedly trains it to convergence, prunes the lowest-magnitude weights, and retrains. This process yields a sparse network that often matches the dense model's accuracy, but at the expense of several times the training cost. We instead ask: can we begin with a sparse network and grow it just enough to match IMP's accuracy, but with far less total training?

To formalize this, let $T$ denote a learning task (e.g., image classification) and $G$ a dense neural network with parameter $\theta \in \mathbb{R}^n$. A sparse subnetwork $G'$ of $G$ is defined by a binary mask $m \in \{0, 1\}^n$ with effective weights $\theta' = m \odot \theta$. Its density is $\rho(G') = \frac{||m||_0}{n}$ where $||m||_0$ counts nonzero entries in $m$.

IMP produces a final sparse network $G^{\text{IMP}}$ of density $\rho_{\text{IMP}}$ that performs nearly as well as $G$. Our goal is to construct, starting from an initially sparse network $G'_0$, a grown network $G'$ with smallest possible density $\rho^*$ such that $P(G') \geq P(G^{\text{IMP}}) - \delta$ for a small tolerance $\delta$, while ensuring that its total training cost $C_{\text{total}}(G') \ll C_{\text{total}}(G^{\text{IMP}})$.

# 3 RELATED WORKS

**Pruning after Training and the Lottery Ticket Hypothesis**   A common approach to sparsity is pruning after training: train a dense model to convergence, remove low-magnitude weights, and optionally fine-tune (Hoefler et al., 2021). While effective for inference, retraining from scratch with the same mask often underperforms, as pruned connections may have been useful during learning (Li et al., 2016). The Lottery Ticket Hypothesis (LTH) (Frankle & Carbin, 2018) reframed this by positing that certain subnetworks ("winning tickets") exist within the initial dense network that, when trained from their original initialization, can match dense accuracy. Iterative Magnitude Pruning (IMP) provided empirical evidence for this claim (Frankle & Carbin, 2018).

**Pruning at Initialization**   Pruning at Initialization aims to bypass the expense of training a dense model by deciding which parameters to keep before training begins. Methods such as SNIP (Lee et al., 2018), GraSP (Wang et al., 2020), SynFlow (Tanaka et al., 2020), and PHEW (Patil & Dovrolis, 2021) evaluate the saliency of each connection at initialization using criteria based on gradients, Hessian approximations, or topological measures. The user pre-specifies the density, and the method immediately prunes to that target.

**Dynamic Sparse Training**   Dynamic Sparse Training keeps the network sparse throughout training but periodically rewires connections to improve learning capacity. Typical DST methods alternate between pruning unimportant weights and adding new ones, guided either by random selection (Mocanu et al., 2018; Mostafa & Wang, 2019) or by data-driven criteria such as gradient magnitude (Liu et al., 2020; Evci et al., 2020), momentum (Dettmers & Zettlemoyer, 2019), or a combination thereof (Heddes et al., 2024). These approaches have also been extended to structured sparsity (Lasby et al., 2024) and have demonstrated robustness against image corruptions (Wu et al., 2025). The target density is fixed from the start — the rewiring steps are designed to maintain this constant sparsity while improving topology over time.

**Optimization-based Sparsification**   Optimization-based sparsification integrates the sparsity mask into the training process, making it a learnable parameter alongside the network weights. Starting from STR (Kusupati et al., 2020), new methods are being proposed to optimize structured sparsity (Yuan et al., 2021), enable more efficient mask updates (Zhang et al., 2022; Yuan et al., 2021; Zhou et al., 2021). Another thread of work fixes the edge weights to be randomly initialized and only optimizes the binary mask (Ramanujan et al., 2020; Wortsman et al., 2020). Like DST, these methods adapt topology over time, but they either specify target density explicitly in advance or incorporate it into the optimization objective (Kusupati et al., 2020; Yuan et al., 2021).

Unlike pruning or DST, PWMPR does not require a preset density: it discovers the density automatically during growth, stopping once accuracy gains plateau

# 4 METHOD

We propose an iterative framework that begins with a sparse network, and iteratively trains and grows the network, until the performance plateaus. Designing this iterative training-and-growth scheme requires answering several methodological questions, such as how to initialize the sparse network, when and where to grow new connections, how many connections to add, how to initialize their weights, and when to stop the process. Among these questions, the most critical is *where to grow new connections*, as it directly determines the network's ability to learn and generalize effectively.

## 4.1 WHERE TO GROW? GROW HIGH WEIGHT PATHS AND AVOID BOTTLENECKS

**Create High-weight Paths for Faster Convergence**   Consider a feed-forward network $f(\cdot, \theta)$ trained by (stochastic) gradient descent. One update step is given by

$$f_{t+1} = f_t - \eta\, \Theta_t\, \nabla_f \mathcal{L}, \qquad \Theta_t = \nabla_\theta f_t\, \nabla_\theta f_t^\top, \tag{1}$$

with $\eta$ the learning rate, $\Theta_t$ the *Neural Tangent Kernel* (NTK), and $\mathcal{L}$ the empirical loss function. Directions whose NTK eigenvalues are large lead to faster convergence (Arora et al., 2019). Gebhart

et al. (2021) express the network output at node $k$ as a sum over paths

$$f^k(x, \theta) = \sum_s \sum_{p \in \mathcal{P}_{s \to k}} \pi_p(\theta) \, a_p(x, \theta) \, x_s, \tag{2}$$

where $\mathcal{P}_{s \to k}$ is the set of paths from input $s$ to $k$, $\pi_p(\theta) = \prod_{(i,j) \in p} \theta_{ij}$ is the path's weight product, and $a_p$ is an activation indicator. Using the chain rule they factor the NTK into a data term and the *Path Kernel* $\Pi_\theta = \nabla_\theta \pi(\theta) \, \nabla_\theta \pi(\theta)^\top$ which depends only on weights and topology. Maximizing the trace of the path kernel accelerates convergence as it governs NTK eigenvalues. With a newly added zero-weight-initialized connection $(i, j)$, the path kernel trace increases by

$$\Delta \operatorname{Tr}(\Pi_\theta)_{(i,j)} = \sum_{p | (i,j) \in p} \Big( \prod_{(u,v) \in p'_{(i,j)}} \theta_{uv} \Big)^2. \tag{3}$$

where $p'_{(i,j)} := p \setminus \{(i,j)\}$. So, adding connections with high $\Delta \operatorname{Tr}(\Pi_\theta)_{(i,j)}$ is expected to speed up convergence. $\Delta \operatorname{Tr}(\Pi_\theta)_{(i,j)}$ is expensive to compute, as it requires enumerating all paths in the network. In this study, we propose a $L_1$ surrogate scoring metric to guide growth.

$$S(i, j) = \sum_{p | (i,j) \in p} \prod_{(u,v) \in p'_{(i,j)}} |\theta_{u,v}| \tag{4}$$

which we term it the *Path Weight Magnitude Product* (PWMP) score. The PWMP score is cheaper to compute for all potential connections at once, and it also has a clear interpretation: it measures the contribution to total weight magnitude product gained by adding per unit of weight to the connection $(i, j)$.

**Avoid Bottlenecks for Better Generalization**    While maximizing path kernel trace can speed up convergence, it does not guarantee good generalization. Prior work (Patil & Dovrolis, 2021) shows that sparse networks that maximize trace under a fixed density tend to collapse into narrow hidden layers, or bottlenecks. Such bottlenecks restrict the diversity of input-output paths and lead to worse generalization compared to broader, more evenly distributed structures.

Layer width alone does not capture bottlenecks, since connections can concentrate on a few nodes. We use the $\tau$-core measure (Batta et al., 2021) to quantify concentration. The idea is to measure how concentrated the network's paths are. For each node $v$, we define its path centrality as the total weight magnitude product of all paths that pass through it:

$$C(v) = \sum_{p : v \in p} \prod_{(i,j) \in p} |\theta_{i,j}| \tag{5}$$

The weighted $\tau$-core is then the smallest set of nodes that together account for at least a fraction $\tau$ (e.g., 90%) of the total path centrality. A small $\tau$-core indicates that only a few nodes dominate the flow of information — a bottleneck — whereas a larger $\tau$-core reflects more balanced path distribution.

**Algorithm: PWMP-biased Random Growth (PWMPR)**    Based on the above insights, we propose a probabilistic growth algorithm, which we call PWMP-biased random growth (PWMPR). As illustrated in Figure 2, we first compute the PWMP score $S(i, j)$ for every potential connection $(i, j)$, and then sample new connections to add with probability proportional to $S(i, j)$. PWMPR avoids bottlenecks by sampling edges in proportion to PWMP, balancing convergence speed with structural diversity.

$S(i, j)$ can be computed efficiently using a single forward and backward pass on a network where we convert every weight to its absolute value. In the forward pass, we feed an all-ones input into the network. The activation of each node $v$ is equal to the total PWMP for all paths from input to $v$, referred to as the complexity of $v$. In the backward pass, we compute the gradient of the output with respect to node $v$'s pre-activation, which is equal to total PWMP for all paths from $v$ to output, referred to as the generality of $v$. With these quantities, the PWMP gain $S(i, j)$ for a potential edge $(i, j)$ is computed as the product of the complexity of node $i$ and the generality of node $j$. Both passes operate in $\mathcal{O}(E)$ time, where $E$ is the existing edge count.

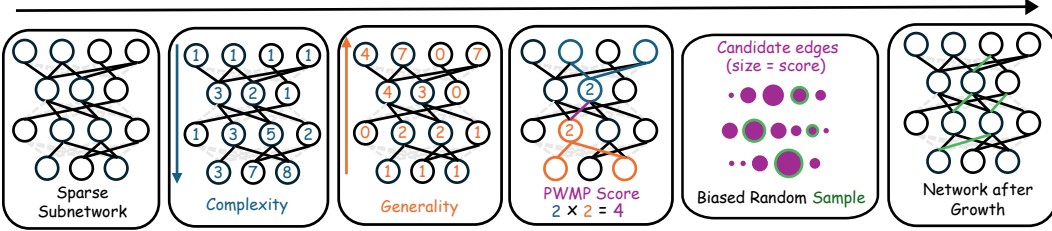

Figure 2: Illustration of the PWMP random growth algorithm, including computation of PWMP scores through forward/backward passes and sampling new connections based on the scores. This illustration uses a simplified network where all edge weights equal to 1.

### 4.2 OTHER METHODOLOGICAL DESIGN CHOICES

**Where do we start? Initialize with PHEW**  We initialize the sparse network using PHEW (Patil & Dovrolis, 2021), which provides a performant starting point at low density. We choose the initial density $\rho_{init}$ to be low enough to allow thorough exploration of the density range, but high enough to avoid disconnected nodes (i.e. nodes with no incoming or outgoing connections). More details on initialization are provided in Appendix A.4.

**When to Grow? Grow Early in the Training Process**  Frankle & Carbin (2018) performs pruning after a number of epochs long enough for the dense network to fully converge (i.e., "extensive training"), to remove weights unimportant for the final learned function. For iterative growth scenarios, however, we hypothesize that growth decisions made only after a few epochs can be as good as those made after extensive training. This hypothesis is motivated by observations that early in training, gradients are stronger, more coherent, and better aligned across updates, providing reliable signals for selecting connections (Jastrzebski et al., 2020; Wang et al., 2020; Dettmers & Zettlemoyer, 2019). As training nears convergence, gradient magnitudes shrink and become noisier (Jastrzebski et al., 2020), reducing their usefulness for guiding new connections. Thus, we adopt a training-and-growth strategy in which growth occurs after only a fraction $\omega$ of the total dense-network training budget, which reduces the cumulative training cost of the iterative process.

**When to Stop? Stop when Early Training Performance Plateaus**  A natural strategy for determining when to stop iterative training and growth is to monitor performance improvements and halt once further increases in density yield diminishing returns. However, our method does not involve extensive training at every density level, so whether the performance after extensive training has plateaued is not directly observable. We hypothesize that performance-density curve during iterative growth process can serve as a proxy for selecting a suitable stopping point.

Different from previous works (Adriaensen et al., 2023; Egele et al., 2024) that focuses on determining stopping point based on a running trajectory with many datapoints, here we need to estimate the plateau point on a complete but sparse set of datapoints. Thus, prpose a simple logistic fit of the performance-density curve to capture the diminishing-return effect and estimate the plateau onset:

$$P(G_k) = P_0 + A \left(1 - e^{-\beta \rho_k}\right)$$

where $\rho_k$ denotes network density, $P(G_k)$ is the observed performance. We estimate parameters $\hat{P}_0$, $\hat{A}$, and $\hat{\beta}$ by minimizing the mean squared error between the model and empirical observations. The plateau onset $\hat{\rho_k}$ is then defined as the smallest density at which the predicted performance reaches 95% of its asymptotic value.

**How Much to Grow? Two-stage Exponential Growth**  At each iteration, we increase density by a fraction of the current density:

$$\Delta\rho(m_k) = \gamma \cdot \rho(m_k),$$

where $\gamma$ is a growth ratio. This proportional strategy results in exponential growth, facilitating efficient exploration of the density-performance landscape. We use $\gamma = 25\%$ in our experiments, matching the pruning ratio of 20% used in IMP (Frankle & Carbin, 2018).

**How to Initialize New Connection? Zero Weights**   Following Evci et al. (2020), we initialize the weights of newly added connections to zero. This avoids introducing noise or interference from random initializations and allows the network to learn appropriate values through gradient descent. Zero initialization also maintains stability in the functional behavior of the current network.

## 5   EXPERIMENTS

**Dataset and Architectures**   We evaluate our method on standard image classification tasks using *CIFAR* (Krizhevsky et al., 2009), *TinyImageNet* (Le & Yang, 2015) and *ImageNet* (Russakovsky et al., 2015) to assess performance from small- to large-scale settings. The corresponding network architectures include *ResNet* (He et al., 2015) and Vision Transformer (*ViT*) (Dosovitskiy et al., 2021). For *ViT*, we adopt scaled-down versions, following prior practices for training transformers on smaller datasets (Lee et al., 2021).

**Pruning Scope**   For *ResNet*, we follow the setup in Frankle & Carbin (2018), leaving the final linear layer and shortcut connections unpruned. For *ViT*, we follow the common practice of keeping the embedding layer and classification head dense, and additionally avoid pruning the query and key projection matrices in the attention modules. Because path-kernel signals do not correlate well with Q/K weights, we keep those dense and defer attention-specific growth to future work.

**Baseline Methods**   We have identified three categories of baseline methods for comparison, each serving a distinct purpose in evaluating our approach. First, we compare against iterative magnitude pruning (IMP) (Frankle & Carbin, 2018) in terms of performance, density and training cost. We use the continued training variant (IMP-C) as our primary benchmark, since it produces stronger sparse networks by relaxing the requirement of retraining from scratch. Second, we evaluate alternative growth strategies, including random growth (Mocanu et al., 2018) (RG) and gradient-based growth (Evci et al., 2020) (GG), to test the effectiveness of our method. Third, we include methods that find good sparse networks given target densities, such as PHEW (Patil & Dovrolis, 2021), RigL (Evci et al., 2020), DSR (Mostafa & Wang, 2019), SparseMomentum (Dettmers & Zettlemoyer, 2019) and GSE (Heddes et al., 2024). While these methods do not directly address our problem of density discovery, we compare performance of networks found at different density levels. Under our iterative training-growth scheme, the total training (in terms of parameter updates) naturally increases with density. To ensure fairness, we scale the number of epochs for PHEW and RigL at each density so that their total training matches our scheme.

## 6   RESULTS

**Early Growth Creates Networks As Good As Growth After Convergence**   How does the timing of growth decisions affect final performance? We study this on *CIFAR-10* and *ResNet-32*. In each iteration of our training-and-growth framework, we apply PWMPR, then perform a short period of training—"rough training"—before the next growth step. We test rough training schedules of 5 epochs, 10 epochs, and an adaptive early-stopping rule that halts if validation loss fails to improve for 3 epochs. After each density increment, we apply an "extensive training" phase to assess the resulting topology. Figure 3 shows the results. As expected, longer rough training improves intermediate accuracy. However, after extensive training, performance differences become very small, especially between 10 epochs and the adaptive rule. Thus, early growth—made after only a fraction $\omega$ of full training—can produce topologies as effective as those grown after convergence. Repeating this experiment with GG yields the same conclusion (Appendix **??**).

**PWMPR Outperforms or Matches Other Growth Methods**   Can PWMPR effectively identify strong sparse topologies? We first compare it with gradient-based growth (GG) and random growth (RG), as shown in Figure 4. On *CIFAR* benchmarks, PWMPR matches or exceeds both baselines. Its advantage becomes clearer on *TinyImageNet / ResNet-18*. For *TinyImageNet / ViT*, PWMPR is slightly weaker than GG below 40% density, but outperforms it at higher densities. A key advantage of PWMPR is efficiency: GG estimates gradients of missing connections by temporarily treating the network as fully connected and processing a batch of examples, incurring high overhead, while PWMPR is purely topological and requires only one forward-backward pass on the sparse network.

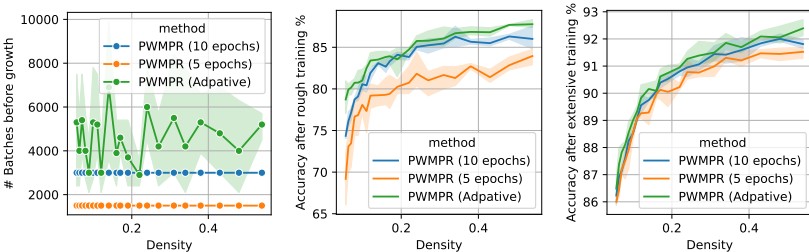

Figure 3: Effect of growth timing on *CIFAR-10* with *ResNet-32*. We compare three schedules: 5 epochs, 10 epochs, and an adaptive early-stopping criterion at each sparsity level. (a) Number of batches before each growth decision. (b) Accuracy after rough training. (c) Accuracy after extensive training.

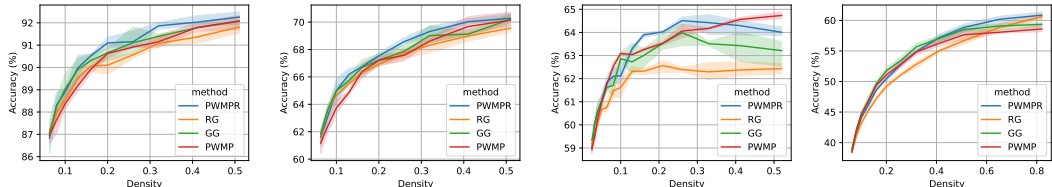

Figure 4: Performance-density relationship of PWMPR compared with other growth mechanisms (RG and GG) and the ablated version (PWMP). (a) *CIFAR-10 / ResNet-32*. (b) *CIFAR-100 / ResNet-56*. (c) *TinyImageNet / ResNet-18*. (d) *TinyImageNet / ViT*. All experiments use the same iterative training-and-growth framework, and thus share the same training budget.

We also assess an ablated variant, PWMP, which deterministically adds connections that maximize total PWMP. PWMP generally underperforms PWMPR, supporting the importance of avoiding bottlenecks.

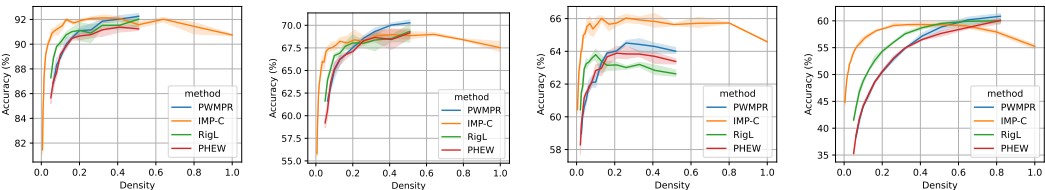

Figure 5: Performance-density relationship of PWMPR compared with other sparsity strategies (IMP-C, RigL, and PHEW). (a) *CIFAR-10 / ResNet-32*. (b) *CIFAR-100 / ResNet-56*. (c) *TinyImageNet / ResNet-18*. (d) *TinyImageNet / ViT*. For IMP-C, we evaluated performance over densities from 100% down to 1%. For PWMPR, PHEW, and RigL, results were collected up to approximately 50% density, for the sake of computational cost. For PHEW and RigL, we scale the number of training epochs at each density to match the cumulative training cost of PWMPR.

**PWMPR Finds Competitive Sparse Topology**   Having established PWMPR as a strong growth-based method, we now examine its ability to identify high-performing sparse topologies. Figure 5 shows performance-density tradeoffs across methods. On *CIFAR* benchmarks, PWMPR finds sub-networks that match or exceed IMP-C performance at densities of roughly 40% and 30%, respectively. These densities are higher than those at which IMP-C retains near-optimal accuracy (∼15% and 20%), reflecting IMP-C's advantage of pruning from a fully trained dense model, which provides richer importance signals. On *TinyImageNet / ViT*, it achieves parity but at around 50% density—likely because the chosen *ViT* architecture (Heo et al., 2021; Lee et al., 2021) is designed for small datasets and has substantially fewer parameters than *ResNet-18*. Figure 5 also compares PWMPR with PHEW and RigL under matched training budgets. PWMPR consistently outperforms

Table 1: Top-1 accuracy on *ImageNet / ResNet-50*; PWMPR results averaged over 3 runs.

| Method | 10% Density | 20% Density |
|---|---|---|
| DSR (Mostafa & Wang, 2019) | 71.6 | 73.3 |
| Sparse Momentum (Dettmers & Zettlemoyer, 2019) | 72.3 | 73.8 |
| RigL (Evci et al., 2020) | $73.0 \pm 0.04$ | $75.1 \pm 0.05$ |
| GSE (Heddes et al., 2024) | $73.2 \pm 0.07$ | N/A |
| **Iterative Growth** | $71.0 \pm 0.04$ | $73.2 \pm 0.13$ |

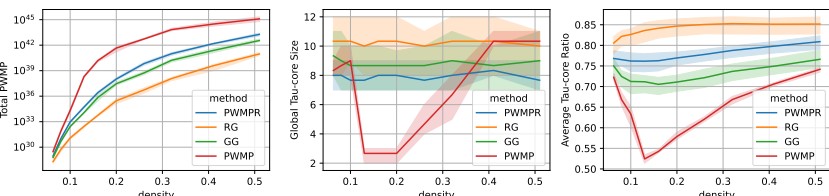

Figure 6: Evolution of topological metrics during iterative growth on *CIFAR-10 / ResNet-32*, comparing different growth mechanisms (PWMPR, RG, GG and PWMP). Left: Total PWMP. Middle: $\tau$-core size ($\tau = 0.9$). Right: Average $\tau$-core ratio (higher = fewer bottlenecks).

PHEW, highlighting the benefit of iterative growth over one-shot pruning at initialization. RigL is competitive at lower densities, but PWMPR surpasses it as density increases.

Finally, we evaluate PWMPR on the large-scale *ImageNet* dataset with *ResNet-50*, comparing against dynamic sparsification methods. Given the high computational cost, we restrict comparisons to reported results at 10% and 20% density. PWMPR begins from a PHEW-initialized network at 2% density and grows with a ratio of $\gamma = 25\%$ until just below the target density, followed by a smaller one-step growth to match the target. Table 1 summarizes the results: On ImageNet, PWMPR lags recent dynamic methods by 2%, reflecting the advantage of connection reallocation, but shows that growth-only density discovery remains viable at scale.

**PWMPR Balances High Path Weight Magnitude Product and Low Bottlenecks** Do networks discovered by PWMPR exhibit the topological properties it is designed to encourage? We evaluate two targeted metrics: the total PWMP and the $\tau$-core (with $\tau = 0.9$). Given the layered structure of neural networks, we measure the *average $\tau$-core ratio* (layerwise $\tau$-core size normalized by width, then averaged) in addition to the global $\tau$-core size. We analyze these metrics on *CIFAR-10* with *ResNet-32* as a representative case.

Figure 6 shows how these topological metrics evolve during growth under different strategies. GG achieves a total PWMP comparable to PWMPR, suggesting that high-gradient connections also raise PWMP, but yields much lower average $\tau$-core ratios, indicating bottlenecks. This may explain why PWMPR surpasses GG at higher densities, where generalization is critical. RG yields the highest average $\tau$-core ratio, theoretically favoring generalization, but its low total PWMP—implying slow convergence—leads to worse performance. Removing randomness from PWMPR boosts total PWMP but sharply lowers the average $\tau$-core ratio at low densities; beyond 13%, the ratio recovers, likely because high-PWMP connections are already added and remaining edges distribute more evenly.

A final observation is that global $\tau$-core size is inversely correlated with performance, in contrast to the average $\tau$-core ratio. We attribute this to residual connections in modern architectures such as *ResNet*, which preserve effective pathways even when most connections in a block are pruned. Thus, strong performance can persist despite a small global $\tau$-core, as long as some of the layers are still wide and remain accessible through residual links.

**Iterative Growth is More Efficient than Iterative Pruning** Does the iterative growth framework enable us to more efficiently identify sparse networks than iterative pruning? We compare the performance and cumulative cost of PWMPR at the stopping point, with the performance and cost

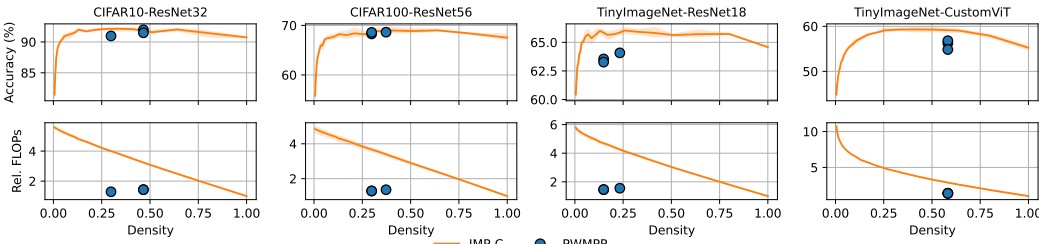

Figure 7: Performance and normalized cumulative cost of PWMPR vs. IMP-C. Each column shows results for a dataset-architecture pair. Top: Accuracy; Bottom: Relative FLOPs (normalized by extensive-dense training FLOPs). IMP-C curves represent results across all intermediate densities. PWMPR is shown in orange dots with each dot representing one of three random seeds.

IMP-C achieves at all intermediate densities, as shown in Figure 7. On both *CIFAR* benchmarks, PWMPR matches IMP-C's performance while requiring only about $1.5\times$ the cost of training a dense model - less than half of the cost of IMP-C. On *TinyImageNet / ResNet-18*, PWMPR again reduces cost by more than half, though with a slight drop in accuracy relative to IMP-C. On *TinyImageNet / ViT*, the efficiency gain is smaller, likely because we use a compact ViT tailored for small datasets and the stopping density is higher ($\sim$58Overall, these results indicate that PWMPR efficiently discovers sparse networks whose performance is competitive with IMP-derived lottery tickets, but at substantially lower training cost.

## 7 DISCUSSION AND LIMITATIONS

We introduced Path Weight Magnitude Product-biased Random growth (PWMPR), a growth-based method for training sparse neural networks that simultaneously constructs subnetworks and discovers their operating density. Unlike pruning approaches that destructively remove connections or dynamic sparse training methods that maintain a fixed sparsity level, PWMPR adopts a constructive sparse-to-dense paradigm: starting from a sparse seed, it selectively grows edges guided by path-weight signals, mitigates bottlenecks through randomized sampling, and stops when accuracy gains plateau. This shift reframes sparse learning not only as an optimization problem but also as an exploration of how networks can grow into high-performing subnetworks.

Our results on CIFAR, TinyImageNet, and ImageNet demonstrate that PWMPR achieves performance close to IMP-derived lottery tickets, though at higher densities, and does so at substantially lower cost ( 1.5x dense training vs. 3-4x for IMP-C). These findings establish growth-based density discovery as a credible alternative to pruning-based strategies and a complementary paradigm in sparse training.

**Limitations** Despite these contributions, our work has several limitations. First, growth-only methods cannot reach the extreme sparsity levels of pruning, since low-importance connections are never explicitly removed. This explains why PWMPR requires higher density than IMP-C to match accuracy. Second, the PWMP heuristic is tailored to feed-forward and convolutional structures; it does not naturally extend to query-key matrices in attention, where magnitudes decouple from functional importance. Third, our stopping rule, based on logistic-fit extrapolation, is a simple heuristic compared to more sophisticated learning-curve extrapolation techniques. Finally, our experiments are limited to vision benchmarks with relatively modest-scale transformers; validation in large-scale NLP or speech domains remains an important direction for future work.

Taken together, these limitations point to natural extensions: hybrid grow-prune methods to combine the benefits of constructive and destructive updates; attention-specific growth rules informed by synaptic diversity or head-level importance; and broader domain validation. More broadly, we view PWMPR not only as a method but as the foundation of a growth-based research agenda for sparse learning — one that complements pruning and DST, and expands the conceptual landscape of how efficient subnetworks can be discovered.

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

## A METHOD DETAILS

### A.1 MATHEMATICAL DERIVATION OF PWMP SCORE

Consider a feed-forward network $f(\cdot, \theta)$ trained by (stochastic) gradient descent. One update step is given by

$$f_{t+1} = f_t - \eta \, \Theta_t \, \nabla_f \mathcal{L}, \qquad \Theta_t = \nabla_\theta f_t \, \nabla_\theta f_t^\top, \tag{6}$$

with $\eta$ the learning rate, $\Theta_t$ the *Neural Tangent Kernel* (NTK), and $\mathcal{L}$ the empirical loss function. Directions whose NTK eigenvalues are large lead to faster convergence (Arora et al., 2019). Gebhart et al. (2021) express the network output at node $k$ as a sum over paths

$$f^k(x, \theta) = \sum_s \sum_{p \in \mathcal{P}_{s \to k}} \pi_p(\theta) \, a_p(x, \theta) \, x_s, \tag{7}$$

where $\mathcal{P}_{s \to k}$ is the set of paths from input $s$ to $k$, $\pi_p(\theta) = \prod_{(i,j) \in p} \theta_{ij}$ is the path's weight product, and $a_p$ is an activation indicator. Using the chain rule they factor the NTK into a data term and the *Path Kernel*

$$\Pi_\theta = \nabla_\theta \pi(\theta) \, \nabla_\theta \pi(\theta)^\top \tag{8}$$

which depends only on weights and topology. The trace of the path kernel can be computed as

$$\text{Tr}(\Pi_\theta) = \sum_p \sum_{(i,j) \in p} \left( \frac{\pi_p(\theta)}{\theta_{ij}} \right)^2, \tag{9}$$

Maximizing the path kernel trace accelerates convergence, because it corresponds to the sum of squared path derivatives, which govern NTK eigenvalues. With a newly added zero-weight-initialized connection $(i, j)$, the path kernel trace increases by

$$\Delta \text{Tr}(\Pi_\theta)_{(i,j)} = \sum_{p \mid (i,j) \in p} \left( \prod_{(u,v) \in p'_{(i,j)}} \theta_{uv} \right)^2. \tag{10}$$

where $p'_{(i,j)} := p \setminus \{(i,j)\}$. So, adding connections with high $\Delta \operatorname{Tr}(\Pi_\theta)_{(i,j)}$ is expected to speed up convergence. $\Delta \operatorname{Tr}(\Pi_\theta)_{(i,j)}$ is expensive to compute, as it requires enumerating all paths in the network. In this study, we propose a $L_1$ surrogate scoring metric to guide growth.

$$S(i,j) = \sum_{p|(i,j)\in p} \prod_{(u,v)\in p'_{(i,j)}} |\theta_{u,v}| \tag{11}$$

which we term it the *Path Weight Magnitude Product* (PWMP) score.

## A.2 PWMPR Algorithm Pseudocode

---
**Algorithm 1** PWMPR: Growing New Connections via Single-Pass PWMP Scoring

---
**Require:** Sparse network $G_k = (V, E_k)$ with weights $\theta$, growth budget $\Delta\rho(m_k)$, number of layers $L$

**Ensure:** Updated network $G_{k+1}$ with $n \cdot \Delta\rho(m_k)$ new edges
1: $n \leftarrow |\theta|, \quad M \leftarrow \lfloor n \cdot \Delta\rho(m_k) \rfloor$           ▷ number of edges to add
2: $\tilde{\theta} \leftarrow |\theta|$           ▷ use absolute weights for PWMP computations
3: **Forward pass (complexity).** Feed an all-ones input and propagate through the *sparse* network using $\tilde{\theta}$ to obtain, at every pre-activation node $v$, its *complexity* $\phi(v)$, i.e., the sum of absolute path-weight products from inputs to $v$.
4: **Backward pass (generality).** Define a scalar readout by summing the final-layer pre-activations and backpropagate a unit signal through the *sparse* network with $\tilde{\theta}$ to obtain, for each node $v$, its *generality* $\psi(v)$, i.e., the sum of absolute path-weight products from $v$ to outputs.
5: **for** $l = 1$ to $L$ **do**
6:     **for** each non-existent edge $(i,j)$ between layer-$l$ input node $i$ and output node $j$ **do**
7:         $S(i,j) \leftarrow \phi(i) \cdot \psi(j)$           ▷ PWMP gain estimate for adding $(i,j)$
8: Normalize scores over all missing edges to probabilities $P(i,j) \propto S(i,j)$.
9: Sample $M$ edges without replacement from the set of missing edges according to $P(i,j)$.
10: Add sampled edges to $E_k$ to obtain $E_{k+1}$ and **initialize all new weights to zero**.
11: **return** $G_{k+1} = (V, E_{k+1})$

---

## A.3 Adapting PWMPR to Convolution and Attention

Thus far, we have introduced PWMP in the context of MLP networks. We extend the concept to convolutional and attention layers, explaining how to quantify each parameter's contribution to the overall PWMP.

**Convolutional Layers** In many vision tasks, the input or output of a convolutional layer is shaped $(H, W, C)$, where $H$ and $W$ are the spatial dimensions and $C$ is the number of channels. For each output pixel at spatial location $(i,j)$ in the $k$-th output channel, the convolution is computed as follows:

$$y_{i,j,k} = \sum_{m=0}^{K_h-1} \sum_{n=0}^{K_w-1} \sum_{c=0}^{C_{in}-1} \theta_{m,n,c,k} \cdot x_{i+m,j+n,c} + b_k \tag{12}$$

where $K_h$ and $K_w$ are the kernels' height and width, and $C_{in}$ and $C_{out}$ are the number of input and output channels. This equation illustrates how each weight $\theta_{m,n,c,k}$ participates in all paths that originate at input, pass through $x_{i+m,j+n,c}$, $\theta_{m,n,c,k}$, $y_{i,j,k}$, to the network's output. Since $\theta_{m,n,c,k}$ is repeatedly used at every spatial location, its PWMP contribution is computed by summing its involvement across all $(i,j)$ positions.

**Attention Layers** Consider an input $X \in \mathbb{R}^{t \times d}$, where $t$ is the number of tokens and $d$ is the embedding dimension. A self-attention module is formulated as:

$$\operatorname{Attention}(Q, K, V) = \operatorname{softmax}\left(\frac{QK^\top}{\sqrt{d_k}}\right) V \tag{13}$$

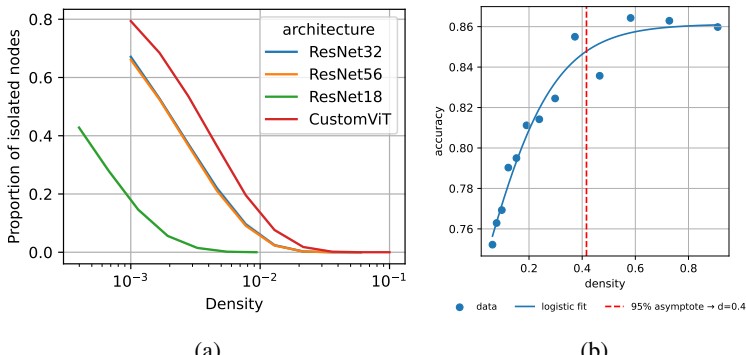

(a)                                                                                                (b)

Figure 8: Illustration of the selection of starting point and stopping point of the iterative growth process. (a) Selection of starting density based on isolated nodes. Fraction of isolated nodes at different density levels. We apply PHEW (Patil & Dovrolis, 2021) to initialize sparse networks and report the proportion of nodes (feature maps in CNNs) that lack both incoming and outgoing connections. (b) Identification of stopping density based on performance-density curves from iterative rough training and growth with *CIFAR-10 / ResNet-32* (single seed). Blue dots: empirical performance; blue curve: logistic fit; red vertical line: predicted stopping point (95% of asymptotic value).

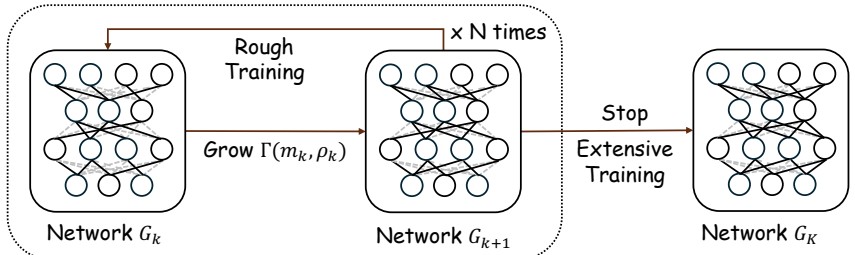

Figure 9: Illustration of the iterative rough training and growth (left to middle), followed by an extensive training at the end (right).

where $Q = X\theta^Q$, $K = X\theta^K$, and $V = X\theta^V$ are the query, key, and value projections of the input, and $\theta^Q$, $\theta^K$, and $\theta^V$ are learnable weight matrices.

This module establishes three distinct computational pathways from the input $X$ to the output. The pathway through $V$ behaves similarly to the weighted sum operation in an MLP. Thus, when adding connections to $\theta^V$ or other feed-forward layers, one can disregard the attention weights—much like activation functions—and focus on identifying connections that maximize the PWMP through the subsequent cascade of linear layers.

In contrast, the pathways through $Q$ and $K$ involve additional matrix multiplications followed by a softmax operation. The softmax normalizes its input, making the resulting attention scores largely insensitive to the absolute magnitudes of $\theta^Q$ and $\theta^K$. As a result, the total PWMP of the network becomes largely decoupled from the magnitudes of these matrices, limiting the proposed algorithm's effectiveness in guiding growth decisions for $\theta^Q$ and $\theta^K$.

Previous work in neural architecture search (Zhou et al., 2022) has shown that different topological metrics are predictive of network performance in attention modules and MLPs within *ViT*. For MLPs, saliency was found to correlate strongly with performance, whereas for multi-head attention modules, a different metric—synaptic diversity (Zhou et al., 2022)—was more effective. To maintain clarity and focus in this study, we keep $\theta^Q$ and $\theta^K$ fixed and defer integration with attention sparsification techniques—such as Jaradat et al. (2024); Lee et al. (2025); Gandhi & Gandhi (2025)—to future work.

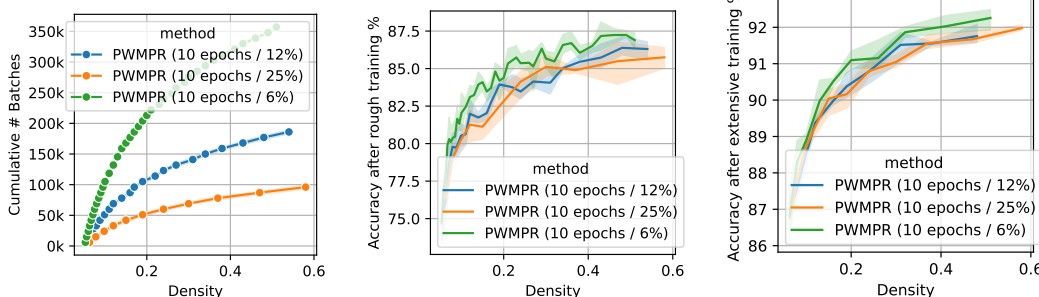

Figure 10: Effect of growth ratio on *CIFAR-10* with *ResNet-32*. We compare three values of $\gamma$: 6%, 12%, and 25%. Missing connections with the highest PWMP score are added (PWMPR). (a) Number of batches before each growth decision. (b) Accuracy just before growth. (c) Accuracy after applying the same extensive training to all models.

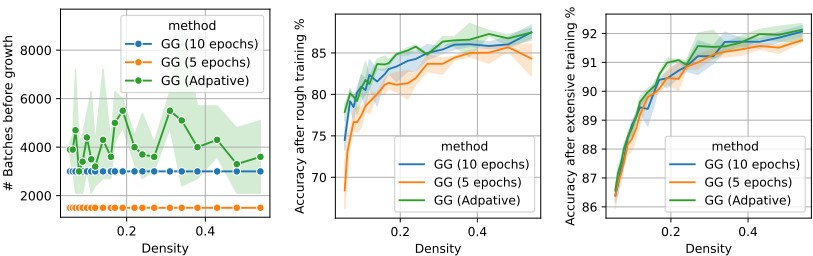

Figure 11: Effect of growth timing on *CIFAR-10* with *ResNet-32*. We compare three schedules: 5 epochs, 10 epochs, and an adaptive early-stopping criterion at each sparsity level. (a) Number of batches before each growth decision. (b) Accuracy after rough training. (c) Accuracy after extensive training.

### A.4  OTHER METHODOLOGICAL DETAILS

**Choosing the Starting Density**  A lower initial density $\rho_{\text{init}}$ allows a more thorough exploration of the entire density range. However, a density that is too low lead to the presence of isolated nodes—i.e., nodes with neither incoming nor outgoing connections. Isolated nodes are problematic, as they cannot receive gradients or be reconnected during growth. Thus, we select initial densities $\rho_{\text{init}}$ high enough to avoid formation of isolated nodes. In Figure 8a, we show the relationship between density and isolated node fraction across different network architectures, which allows us to pick a small $\rho_{\text{init}}$ with no isolated nodes. Table 2 lists the selected $\rho_{\text{init}}$ values.

**Choosing the Growth Ratio**  Regarding the choice of growth ratio $\gamma$, we performed a sensitivity analysis using *CIFAR-10 / ResNet-32*, as shown in Figure 10. We experimented with three values of $\gamma$: 6%, 12%, and 25%. With each setting, we applied PWMPR to grow the network iteratively from an initial density of 0.5% up to 50%. We found that a smaller $\gamma$ leads to finer growth steps, which lead to better performance of the network at each growth step, as shown in 10. Nevertheless, smaller $\gamma$ also leads to more growth steps, which increases the overall training cost.

**Additional Experiment on Gradient-based Growth to Support Early Growth**  In addition to the main experiments on growth timing, we conducted an additional experiment to evaluate the performance of gradient-based growth (GG) with different growth timings. Under the same experimental setup, similar observations were made that early growth (e.g., after 10 epochs) can produce topologies that match the performance of those grown after full convergence, once fully trained.

| Dataset | Architecture | Weights | $\rho_{\text{init}}$ | Iterations/Batch | Optimizers |
|---------|--------------|---------|------|------------------|------------|
| CIFAR-10 | ResNet-32 | 460K | 5% | 30K / 128 | SGD 0.1 |
| CIFAR-100 | ResNet-56 | 850K | 5% | 30K / 128 | SGD 0.1 |
| TinyImageNet | ResNet-18 | 11.1M | 2% | 70K / 128 | SGD 0.2 |
| TinyImageNet | ViT | 2.8M | 5% | 70K / 256 | AdamW 3e-3 |
| ImageNet | ResNet-50 | 25.6M | 2% | 500K / 256 | SGD 0.1 |

Table 2: Datasets and architectures evaluated in this study. Learning hyperparameters for *CIFAR-10/100* are taken from (Frankle & Carbin, 2018), and for TinyImageNet from (Gani et al., 2022; Patil & Dovrolis, 2021). For *ResNet* experiments, the learning rate is decayed by a factor of 10 at 20K and 25K training iterations. For *ViT* experiments, the learning rate follows a linear warmup schedule until 70K iterations, followed by cosine annealing.

## B  EXPERIMENT DETAILS

**Evaluation Metrics**  For the performance metric, given a network $G$ we evaluate its performance $P(G)$ using classification accuracy. To estimate the cumulative training cost $C_{total}$, we compute the number of floating-point operations (FLOPs) incurred during the iterative processes. This estimate accounts for all multiplications and additions performed during inference, adjusted for the network's sparsity. We exclude the cost of non-linear operations such as activation functions and softmax, as they contribute only a small fraction to the overall computational cost.

**Training Hyperparameters**  All networks are trained from scratch in a supervised manner. The learning hyperparameters used for each architecture-dataset pair are summarized in Figure 2. Following the protocol in (Frankle & Carbin, 2018), we reserve 10% of the training data for validation and train the networks on the remaining 90%. For the *ViT*, we apply the same data augmentation strategies as in (Lee et al., 2021) to improve training stability and performance on smaller datasets.

**Baseline Methods Details**  Iterative magnitude pruning (IMP) alternates between training the network and pruning a fraction of its weights. Specifically, in each pruning cycle, the network is trained and then pruned by removing $p_{imp}$ of the lowest-magnitude weights. In the original lottery ticket hypothesis setup (Frankle & Carbin, 2018), the surviving weights are reset to their initial values while retaining the pruning mask. This procedure is designed to identify "lottery tickets"—subnetworks that, when trained from their original initialization, can match the performance of the full dense network. However, since our goal is to find sparse networks that achieve high performance regardless of initialization, we include a variant of IMP that does not reset weights after pruning. We refer to this variant as *IMP-C* (Iterative Magnitude Pruning with Continued training), which is also known as *Gradual Magnitude Pruning (GMP)* in prior work (Lee et al., 2024).

Gradient-based growth (GG) and random growth (RG) are implemented as described in (Evci et al., 2020) and (Mocanu et al., 2018), respectively. For computation of gradients in GG, we regard all missing connections as connections with a weight of zero, and perform forward and backward passes with one batch of training data.

RigL (Evci et al., 2020) is a dynamic sparse training method that alternates magnitude-based pruning with gradient-guided growth. We follow the original setup, performing updates every 100 iterations with a 20% growth ratio, and using the same ERK initialization, update schedule, and drop/grow criteria.

PHEW (Patil & Dovrolis, 2021) prunes at initialization by constructing high-weight paths via biased random walks. Starting from a randomly initialized dense network, it samples paths in both forward and backward directions and retains the traversed connections, removing all others. While originally designed for feed-forward networks, we adapt PHEW to *ViT* by applying it to the feed-forward blocks and to the value and projection matrices in the attention modules.

## C    DISCLOSURE OF USAGE OF LLM

In the preparation of this manuscript, we use GPT-5 to help polish the writing of the paper based on our original draft. In addition, we use GPT-5 to enrich our discovery of the related works, on top of our own knowledge and literature search. We have verified the correctness of all the content in the paper.

