# OpenReview forum: "Growing Winning Subnetworks, Not Pruning Them: A Paradigm for Density Discovery in Sparse Neural Networks"
_ICLR.cc/2026/Conference — ICLR 2026 Conference Withdrawn Submission_

### Official Review · Reviewer_3umZ · 2025-10-30

**Soundness:** 3
**Presentation:** 2
**Contribution:** 2
**Rating:** 2
**Confidence:** 3

**Summary:**

This paper introduces a Pruning at initialization method based on growing path based networks in combination with random growth to enhance connectivity. The proposed method PWMPR, seeks paths along the largest weights for better trainability. Further the authors introduce a growth method which is able to find the ‘operating’ sparsity of a network, aimed at finding sparsites at which dense performance can still be matched.

**Strengths:**

1. The path based method is simple to compute with a single forward pass to obtain the path kernel.
2. The growth strategy requires less FLOPs than iterative methods like IMP.

**Weaknesses:**

Although this work introduces a new method, a large part of the observations and insights mentioned in this paper have been explored before for PaI.

1. A similar work that takes into account the paths as well as the connectivity is the Node Path Balancing method introduced by [1, 2]. This method also outperforms PHEW and has a similar motivation.
2. The regrow process does not seem to provide additional benefits in terms of training, like RiGL, likely because it is data-agnostic. In this case, is there a difference between growing gradually or starting at a target density and training longer? Does the growing help match IMP better, it seems that in any case the proposed method cannot perform IMP and is only able to match it on the smaller CIFAR datasets, which are easier to optimize.
3. The authors mention in the abstract that the growth helps finding the operating density of the network, but how does this hold in practice. Does the performance drop significantly below this density? How would this compare with other PaI methods which cannot identify this operating density and would this imply that performance would always be worse in extreme high sparsity regimes?
4. Figure 4 shows improvements over gradient based growth and random growth, but the improvements are only up to 20% density, at lower densities the they seem to converge, suggesting topology alone is not enough (this has also been studied by other works [3])
5. Although experimental results show that performance can be matched with IMP for smaller CIFAR datasets, the known limitation of PaI remains as the authors show in Table 1, when compared with DST or other methods.

Overall the proposed method seems to share a large overlap with previous work and the benefits of the proposed growing mechanism are not entirely clear.

[1] Pham, Hoang, et al. "Towards data-agnostic pruning at initialization: what makes a good sparse mask?." Advances in Neural Information Processing Systems 36 (2023)

[2] Xiang, Lichuan, et al. "DPaI: Differentiable Pruning at Initialization with Node-Path Balance Principle." The Thirteenth International Conference on Learning Representations.

[3] Frankle, Jonathan, et al. "Pruning Neural Networks at Initialization: Why Are We Missing the Mark?." International Conference on Learning Representations.

**Questions:**

See above.

---

### Official Review · Reviewer_uZmn · 2025-10-31

**Soundness:** 2
**Presentation:** 3
**Contribution:** 2
**Rating:** 4
**Confidence:** 2

**Summary:**

The paper proposes a new method for training neural networks from sparse to dense efficiently. Inspired by previous analyses on the neural tangent kernel, the paper introduces a new quantity called PWMP score, which is expected to represent how the new connection leads to speed up in convergence. Then the proposed method randomly yields new connections in proportion to the PWMP scores, which is expected to contribute to avoidance of collapsing neurons, with a bunch of heuristic techniques. Experimental results observe various aspects of the proposed method, compared to a variant of IMP and other growth methods.

**Strengths:**

- The derivation of the proposed method is novel. It is inspired by the theory of NTK, following the previous work by Patil and Dovrolis, which enables us to calculate the importance score for each connection.
- In experiments, the paper provides extensive analyses from various perspectives, supporting the empirical benefits of the proposed heuristic techniques.

**Weaknesses:**

- The proposed method consists of a bunch of heuristic techniques, with no theoretical guarantee. The derivation of the PWMP score is based on the previous analyses of NTK, but involves a non-negligible gap from eq (3) to eq (4), which makes unclear how well the proposed method works according to the theoretical insights.
- In Introduction, the paper claims that `We see this as a first step in establishing growth-based density discovery as a new
paradigm in sparse neural network training`. However, the sparse-to-growth approach with unknown density is not a new paradigm. For example, see [Evci et al. 2022] and the papers citing it. The proposed method should also be compared with these existing sparse-to-growth methods in experiments.
- The paper also focuses on the question asking `can we begin with a sparse network and
grow it just enough to match IMP’s accuracy, but with far less total training?` as described in the Problem Statement section. However, from Figure 7, it can be seen that the proposed method cannot achieve the highest accuracy of IMP and the training cost seems on par under the same accuracy, particularly on TinyImagenet.
- The results of Table 1 show that the benefit of the proposed method is reduced on a large-scale dataset. The scalability is a crucial perspective in deep learning, and thus the results imply that the proposed method is less useful than other growth methods.


[Evci et al. 2022] Gradmax: Growing neural networks using gradient information (ICLR'22)

**Questions:**

See weaknesses.

---

### Official Review · Reviewer_egtZ · 2025-10-31

**Soundness:** 2
**Presentation:** 4
**Contribution:** 3
**Rating:** 4
**Confidence:** 5

**Summary:**

The authors propose a novel dynamic sparse training (DST) method PWMPR that progressively grows weights, unlike existing DST methods that either progressively prune weights or jointly grow/prune while maintaining a constant sparsity level. The authors identify a novel growing criteria (PWMP) derived from an analytical analysis using the neural tangent kernel (NTK), with the objective of maximizing convergence. The model is initialized using an existing initialization (PHEW), and with a high-enough initial density (1.0 - sparsity) hyper-parameter to avoid disconnected neurons. The authors propose to cease growing/training when the generalization v.s. density has plateaued, i.e. early stopping, and use a proxy metric to evaluate this. The method is compared with existing DST and Iterative Magnitude Pruning (IMP) baselines on CIFAR, TinyImageNet, and ImageNet with ResNets and Vision Transformers.

**Strengths:**

* Growing neural networks rather than pruning them is an extremely interesting and appealing research direction within the sparse training community with relatively little existing literature. I personally find the high-level motivation of this paper and methodology very compelling.
* Although necessarily many of the criteria for growing and pruning are often heuristic in sparsity, the authors do an excellent job of motivating their growth criteria through analysis, and initialization from prior work.
* The models and datasets on the whole appear to be appropriate for the evaluation.
* The baseline methods are appropriate, if not the setting/usage as described in weaknesses.
* The authors explain many of the weaknesses of their method well, and deserve a lot of credit for this.

**Weaknesses:**

* The method proposed is only evaluated at relatively low sparsity levels of 80 and 90% (i.e. density of 20 and 10%), which makes it hard to compare with sparse training or pruning methods which often work in the realm of >90%.
* The generalization results are nowhere near competitive with existing pruning or sparse training methods, which both operate at higher sparsity (i.e. lower density), and achieve better generalization than the proposed method. The authors argue that it is scientifically interesting regardless, and I might agree if they were in the same ballpark, but these results are very far away from the state of the art in pruning/sparse training.
* At such low sparsities, the experiments and motivation comparing training/convergence times of the proposed method with existing DST methods are not an apple-to-apples comparison: at low sparsities, DST methods can also converge faster than at high sparsities.
* On a similar note, page 6, "total training naturally increases with density" for PWMPR, and so the training epochs are increased with density (i.e. decreased with sparsity). However, RigL/SET require increased training epochs at higher sparsity (i.e. lower density), and so the DST baselines are being put at a disadvantage in this setup. Rather, if generalization is the key factor then there should be a hyperparam sweep over training epochs to select the best.
* Results should be compared at a density/sparsity that makes sense given real-world efficiency on real hardware (i.e. >95%).
* There are abstract claims of efficiency in section 1 and page 9 e.g. (1.5x dense) with no timings real-world hardware. No explanation or justification as to how "growing" weights is more or even equally efficient on real-world hardware for training, as although it might naively seem it should be, any savings in compute in early training would easily be surpassed by memory allocation overheads especially on GPUs/accelerators, making this less efficient than starting with a higher density and pruning or even potentially dense training.
* Assuming the method was learning to grow representations in a meaningful way, I'd hope to see different final density/sparsity levels for different problems or datasets, but this is not happening as far as I can see.
* Claims around faster convergence only compare to IMP, but don't compare to DST baselines, which are much more appropriate.

Minor comments:
* The authors motivate the method ad ask in the problem statement if they can match IMP's accuracy, but I believe DST is the more appropriate comparison given both the motivation and methodology.
* Missing Appendix reference on page 6 at the end of the first paragraph for section 6.
* PWMPR is a hard to remember, spell or pronounce acronym, would highly recommend coming up with something else more memorable.
* While it's possible density might be easier to think of in the context of growing, I really think it's more appropriate and transparent to use sparsity in the paper as this is what the other literature defaults to, and it makes the comparisons slightly more transparent.

**Questions:**

* What are the results at higher sparsity (lower density) levels, while I believe it's obvious from the 90% sparsity results that DST is doing better, I am curious from a scientific standpoint how quickly the generalization decreases > 90%.
* What is the real-world training cost of the method compared to baselines such as SET, RigL and Dense training at the same sparsity? Do you have real-world timings to back the efficiency claims on GPUs/accelerators?
* If you use PWMPR in a setting where you would expect to need more capacity to learn (e.g. self-supervised approach like SimCLR where you are learning instance-level representations vs. class-level for ResNet-50/ImageNet), would you expect it to learn to grow more weights? Is there any evidence this is happening?
* What other growth and initialization criteria did you explore, and how poorly did they work compare to the final methodology? (again mostly scientific curiosity here, although an ablation would be nice).
* What is the relative convergence of RigL/SET compared to PWMPR at 80% and 90% sparsity when using the same numbers of training epochs?

---

### Official Review · Reviewer_Num8 · 2025-11-01

**Soundness:** 2
**Presentation:** 2
**Contribution:** 2
**Rating:** 4
**Confidence:** 4

**Summary:**

This paper explores LTH from sparse to dense direction, instead of pruning a complete dense network. It uses a network grow strategy to explore this problem and validates the method effectiveness on several benchmark datasets.

**Strengths:**

1. LTH is still a open research problem which hasn't been solved yet, then this paper explores a valuable topic.
2. Studying from sparse to dense is a different way to explore compared with previous approaches which prune a dense model to find the winning ticket.

**Weaknesses:**

1. Literature review is not sufficient, there are a lot of LTH related research works, but missed by this draft.
2. The goal is confusing to me. This paper is more like a dynamic sparse training exploration, but it is placed under the LTH topic, which is not clear to show the motivation and conclusion of this paper.
3. If I understand this draft from LTH perspective, the proposed method should be compared with other LTH baselines, which are missed. If I understand it from dynamic sparse training aspect, I am confused why this is under LTH scenario, and the used method looks like a commonly used sparse training strategy with growing or removing operation.
4. The term density appears frequently in the draft, it causes some confusions, does it mean sparsity? or sparsity ratio? Like the title says, discovering density, what does it refer to?
5. Figures and tables need polish. Figure content is hard to follow and the author may want to re-arrange the font size and make they match with each other from the size aspect.

**Questions:**

Please check above section.

---

### Note · Authors · 2025-11-14

I have read and agree with the venue's withdrawal policy on behalf of myself and my co-authors.